# Addressing food insecurity in early childhood programs through a health equity lens: A qualitative case study of Brazil's *Criança Feliz* program

Gabriela Buccini[1], Poliana de Araújo Palmeira[2]*, Ana Poblacion[3], Muriel Bauermann Gubert[1,4]*

1 Department of Social and Behavioral Health, University of Nevada, Las Vegas, Nevada, United States of America, 2 Centro de Educação e Saúde, Universidade Federal de Campina Grande, Cuité, Paraíba, Brasil, 3 Boston Medical Center, Boston, Massachusetts, United States of America, 4 Departamento de Nutrição, Universidade de Brasília, Brasília, Distrito Federal, Brasil

* gabriela.buccini@unlv.edu

## Abstract

### Background

Food insecurity negatively impacts early childhood health and development. Recommendations for establishing equitable actions to address food insecurity in early childhood programming have not been developed globally. The Brazilian *Criança Feliz* Program (PCF), one of the largest early childhood programs worldwide, has faced implementation challenges while addressing food insecurity. We aimed to (i) understand how food insecurity affects the PCF's internal (i.e., organizational-level factors) and external (i.e., family-level and system-level factors) implementation contexts, and (ii) develop equity-focused recommendations for early childhood programs to mitigate food insecurity.

### Methods

Qualitative case study analyses of in-depth interviews with PCF implementation teams and families in five Brazilian municipalities. Participants shared their experience with PCF implementation quality, including questions related to food insecurity. A three-stage rapid qualitative approach was used: (a) inductive thematic analysis identified central codes related to food insecurity, (b) deductive approach to organize central codes within themes related to internal and external contexts of PCF operations, and (c) integration of findings into a set of equity-focused recommendations based on the four categories of the Getting to Equity (GTE) framework.

**Data availability statement:** A de-identified data set is not possible to provide due to ethical and legal considerations. These sharing restrictions are imposed by the UNLV Institutional Board Review (IRB). The authors declare that a de-identified data set from this study are available upon request directly to the UNLV IRB (irb@unlv.edu).

**Funding:** The research was supported by "The Eunice Kennedy Shriver National Institute Of Child Health & Human Development of the National Institutes of Health" under Award Number R00HD097301 (PI: Buccini). The content is the sole responsibility of the authors and does not necessarily represent the official opinion of the National Institutes of Health. The funders had no role in study design, data collection and analysis, decision to publish, or preparation of the manuscript.

**Competing interests:** The authors have declared that no competing interests exist.

## Results

240 interviews were conducted. Internal programmatic barriers included lack of protocols for screening, referring, and following up with families struggling to access food as well as challenges to engage them in early learning activities. External programmatic barriers included family-level factors (e.g., unrealistic support expected from PCF) and system-level factors (e.g., bureaucracy in accessing safety nets). Ten equity-focused recommendations across GTE framework focused on improving program curriculum and protocols to mitigate food insecurity and increasing individual and community capacity.

## Conclusions

We documented barriers at the family, program, and system levels to address food insecurity in the Brazilian PCF home visiting program. Barriers informed the generation of equity-focused programming recommendations to improve practices to address food insecurity, not only for the PCF, but also for the global community implementing home visiting programs.

## Background

Food insecurity is characterized by the lack of regular access to enough safe and nutritious food for normal growth and development and an active and healthy life, due to a lack of resources to obtain food [1,2]. About 30 percent of the global population – 2.4 billion people – were food insecure in 2022, making it a major public health issue [3]. To achieve food and nutrition security, as outlined in the 2030 Sustainable Development Goals, four interrelated pillars must be fulfilled: **availability** – food must be available; **access** – households must have access to it; **utilization** – households must utilize it appropriately; and **stability** – the whole system must be stable. Due to the complex factors contributing to food insecurity, the Global Strategic Framework for Food Security and Nutrition was established to assist countries in developing multisectoral strategies, policies and actions with the goal of improving the four pillars of food and nutrition security by 2030 [4].

In Brazil, the largest country in Latin America and the Caribbean with more than 215 million inhabitants, 27.6% of households were food insecure in 2023 [5]. Households with children were disproportionately impacted by food insecurity with 66.1% of households experiencing food insecurity and 18.1% severe food insecurity, in which children were hungry [5,6]. The presence, severity, and persistence of food insecurity has been associated with poor early childhood development, including delays in the academic, cognitive, behavioral, and socioemotional domains [7–9]. The association between food insecurity and early childhood development outcomes can occur through at least two pathways. The nutritional path occurs through inadequate nutrition whether by the lack of adequate quantity or low-quality of food with negative implications for child development. The psychosocial and emotional paths,

i.e., caregiver worry and anxiety trigger stress-related hypothalamic-pituitary axis and can affect parents psychologically impairing parenting practices (e.g., harsh discipline) with negative implications for child development [7,8,10–14]. The latter path can act as both consequence or mechanism exacerbating stress leading to higher risk of food insecurity. Therefore, food insecurity is a sensitive measure of inequities in early childhood development contributing to the perpetuation of the cycle of poverty [7].

To protect the approximately 10 million young Brazilian children living in extreme poverty, the national government created the *Criança Feliz* Program (PCF) [15]. The PCF seeks to provide (1) home visits based on the Care for Child Development curriculum to foster child stimulation and responsive parenting skills [16], and (2) complementary multisectoral actions to mitigate socio-vulnerabilities of participating families. The PCF uses a multilevel implementation strategy across the three administrative levels of the Brazilian government. The national level is responsible for funding, articulating the multisectoral approach, and coordinating the implementation by developing protocols, training, and monitoring strategies. The state level provides technical support to municipal teams. At the municipal level, teams are responsible for home visits and complementary multisectoral actions [17,18] (see S1 Table for detailed program description).

In 2022, PCF surpassed 57 million home visits delivered to support parenting and early childhood development across 3,028 Brazilian municipalities (out of 5, 570 municipalities) [15]. It became one of the largest early childhoods program globally guided by the five components of the Nurturing Care Framework (i.e., health, nutrition, early learning, security and safety, and responsive caregiving). Accordingly, the Nurturing Care Framework components have proven to be a useful roadmap to design a multisectoral approach to address food insecurity in the context of early childhood development [19]. Yet, several challenges in operationalizing the multisectoral approach have been documented hindering the ability of PCF to address social determinants of health (i.e., re non-medical factors that affect health outcomes) such as food insecurity [18,20].

In 2023, after PCF reached its full implementation, the program was incorporated into the Brazilian Unified Social Protection System [21]. Thus, opportunities arising to optimize the multisectoral nurturing care approach to address food insecurity among households with young children enrolled in PCF will likely yield large social returns. Evidence-informed recommendations for establishing equitable actions to address food insecurity in early childhood programming need to be developed for Brazil and lessons learned shared globally. The Getting to Equity (GTE) is a public health nutrition framework used to guide practitioners and researchers in designing recommendations to promote equity and increase the impact of food security policies, systems, and programs [22–25]. However, to our knowledge, no study has applied the GTE framework to develop equity-focused recommendations for addressing food insecurity in early childhood programs. We aimed to aimed to (i) understand how food insecurity affects the PCF's internal (i.e., organizational-level factors) and external (i.e., family-level and system-level factors) implementation contexts, and (ii) integrate these findings into equity-focused recommendations for early childhood programs to mitigate food insecurity.

## Methods

This study received Institutional Review Board (IRB) approval from the São Paulo State Health Department (CAAE 12872419.6.0000.5469) and by the IRB at University of Nevada, Las Vegas (n. 1702327). The research committees of the participating municipalities and departments granted additional approvals. All participants provided verbal informed consent. The Consolidated Criteria for Reporting Qualitative Studies (COREQ) [26] was followed to report results of this manuscript (S2 Table).

### Study design

This exploratory qualitative case study [27] analyzed in-depth interviews with implementation teams and families participating in PCF to understand how food insecurity impacts the PCF internal (i.e., organizational-level factors) and external (i.e., family-level and system-level factors) implementation contexts. In-depth interview is an appropriate to provide both

breadth and depth of understanding of programming and implementation barriers. Due to the exponential increase of food insecurity after the COVID-19 pandemic, interview guides intentionally included questions about food insecurity and nurturing care multisectoral approach to mitigate food insecurity. Responses to these questions informed data analysis presented in this manuscript.

## Selection of participants

**Municipality recruitment.**  Purposive sampling was used to select information-rich municipality cases for the most effective use of limited evaluation resources [28]. Inclusion criteria to select five Brazilian municipalities were (1) population size, (2) implementation length, (3) implementation model, and (4) geographical region. Four large urban municipalities of Campo Grande-MS, Brasília-DF, Fortaleza-CE, and São Paulo-SP were selected predominantly due to the challenges of implementing PCF in large urban centers [18], and one small predominantly rural municipality (Cuité-PB) was selected for analytical comparison. These municipalities are situated in the North/Northeast (n = 2), Central-west (n = 2), and South/Southeast (n = 1), covering all Brazilian geographical regions.

**Implementation team's recruitment.**  In each municipality, a purposive sampling approach was utilized to identify participants were teams implementing PCF (i.e., home visitors, supervisors, municipal managers, and managers from different sectors interacting with PCF) and families participating in the PCF, both for at least 6 months. The sample size was designed to reach thematic saturation per municipality; therefore, in each municipality, families with different characteristics (i.e., pregnant, caregivers of children under 36 months, child beneficiaries of Cash Transfer Programs) were intentionally interviewed. In-depth interviews were conducted virtually (with or without video according to the participant preference) or via phone with a sample of (a) municipal PCF teams, including municipal coordinators, supervisors, and home visitors (N = 132); (b) municipal managers from other sectors working on the implementation of the PCF, such as health, education, and social assistance (N = 17); and (c) families participating in the PCF (N = 95) (participant characteristics detailed in S3 and S4 Tables). No participants refused to participate. The data were collected from June 2021 to May 2022.

**Method of approach.**  The research team hosted a virtual meeting with each municipality to introduce the study and establish the protocols for scheduling virtual or telephone interviews. For selected families, the supervisor and home visitor assigned to each family were contacted and granted authorization from the family to share their contact information with the research team. A research assistant contacted the families by phone to explain the study's purpose and interview procedures, as well as share a virtual copy of the consent form. If the participant agreed to be interviewed, the research assistant scheduled a convenient time for the interview. A similar approach was followed with PCF teams and municipal managers from other sectors working with PCF.

## Data collection

**Interview guides.**  Interview guides were developed to capture factors influencing PCF implementation contexts. In the interview guide for PCF team and managers, questions followed the RE-AIM Framework dimensions, i.e., Reach, Effectiveness, Adoption, Implementation, Maintenance [29,30]. Interview guides were customized to address the specific needs of different program audiences, resulting in two distinct versions pre-tested with a member of the PCF municipal team and a family member. Pertinent to this study, specific questions about programming approaches to address food insecurity were taken from the implementation dimension. In the interview guide for families, questions followed the Nurturing Care Framework dimensions [31]. Relevant to this study, specific questions about food insecurity and participation in social protection programs in addition to the PCF were taken from the adequate nutrition dimension (S5 Table). Both interview guides had a section on sociodemographic characteristics of participants, which included the validated two-item screening tool (TRIA) to identify Brazilian households at risk for food insecurity (sensitivity of 79.31%,

specificity of 92.95% [32]. Specifically, TRIA uses questions 2 and 4 of the Brazilian Food Insecurity Scale (EBIA): question number 2 in the EBIA (*"Nos últimos três meses a comida acabou antes que você tivesse mais dinheiro para comprar mais?"*) corresponds to HFSSM question Q2: "In the past 12 months, the food that (I/we) bought just didn't last, and (I/we) didn't have money to get more?"); and question number 4 in the EBIA (*"Nos últimos três meses, você teve que se arranjar com apenas alguns alimentos para alimentar os moradores com menos de 18 anos, porque o dinheiro acabou?"*) corresponds to HFSSM question Q4: "In the past 12 months, did (you or other adults in your household) ever cut the size of your meals or skip meals because there wasn't enough money for food?".When a participant caregiver responded affirmatively to both items of the instrument, their household was considered at risk for food insecurity [32].

**Research team's positionality.** Interviews were conducted in Portuguese by trained native Portuguese speaker female researchers with PhD training (GB, MBG, PPS). Data analysis of the food insecurity component of interviews was conducted by female faculty members and co-authors trained in public health nutrition and with experience in research on food insecurity and early childhood in Brazil and the U.S. They all had prior experience conducting qualitative interviews and had no prior relationship with the participants. Participants were told the goals of the study prior to the interview.

**Data management.** In-depth interviews were conducted throughout the months of June 2021 and May 2022. Debriefing meetings were held at the end of each day of interviews to expand field notes and conduct preliminary data coding and analysis. Through this approach, when thematic saturation was achieved in each municipality, the sample size was deemed sufficient for that municipality [33]. Interviews lasted 40–70 min each, were audio recorded with permission after verbal consent was granted, and transcribed verbatim by a professional Portuguese-speaking service. Original audio recordings were compared to transcripts to ensure accuracy.

### Data analysis

**Analytical framework.** The GTE framework was used in this study to integrate qualitative data analysis findings. The GTE framework was first published in 2018 [23], and applied specifically to food insecurity in 2020 [24]. It provides solution-oriented guidance aimed at reducing health disparities, and to eliminate structural factors that inhibit food and nutrition security for all, especially low-income children and families [24]. The GTE has four categories (1) increase healthy options, (2) reduce deterrents, (3) improve social and economic resources, and (4) build on community capacity that help translate the intention to achieve equity into action.

In this study, the premise for using GTE is that disparities in early childhood development cannot be overcome without addressing underlying inequities that cause food insecurity [7–9]. Therefore, in our study, food insecurity is not considered the root cause of existing inequities rather is a metric used to understand who was most affected by the root causes (e.g., poverty). Thus, inequities that GTE informed-recommendations are addressing contribute to food security as a consequential issue.

**Thematic analysis approach.** Rapid qualitative methods, defined as research designed to be responsive and adaptable to changes in context while the study was ongoing [34]. This methodology has been widely used in the field of implementation science to expedite knowledge translation and inform policy decisions [34,36]. A three stage thematic analysis approach was used to analyze data (Fig 1) [35].

In the first stage of analysis, two researchers listened to interviews' audios and recorded information in a predefined matrix to capture themes related to food insecurity throughout PCF implementation. A third researcher compared and validated the information recorded. Questions were discussed and confirmed throughout revisiting original audios. Then, an open-coding process [35] was used to inductively label strategies to addressing food insecurity. During this stage, frequency, complementarity, expansion, and divergence of coded information were considered in order to identify central codes related to food insecurity at family, organizational, and system levels [35,36]. Several central codes related to food insecurity emerged as challenges to the implementation quality of the PCF [18] (Fig 1). The preliminary findings of the first stage of thematic analysis were presented to each municipality for feedback, fostering reflection between implementers

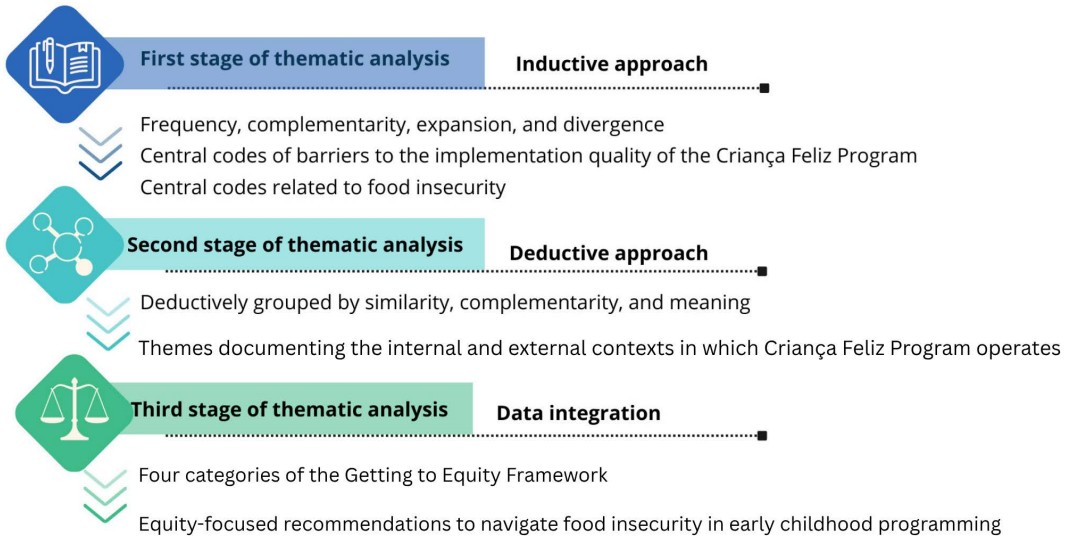

**Fig 1. Stages of data analysis.**

and the research team. Insights from this interaction generated an important research question of how the high prevalence of food insecurity among participants influence PCF implementation at family, organizational, and system levels. This question guided the second stage of thematic analysis as well as the integration of data included in this manuscript.

In the second of analysis, in-depth analyses were conducted to understand how food insecurity impacts the internal and external contexts of PCF operations. Thus, the central codes related to food insecurity identified in phase 1 of data analysis were deductively grouped into two major themes by similarity, complementarity, and meaning as follows: (1) internal context in which PCF operates (i.e., organizational-level factors such as program curriculum) and (2) external context in which PCF operates (i.e., family-level and system-level factors). Quotes from interviews that best represented themes were selected to illustrate the results and were translated from Portuguese (Fig 1).

In the third stage of analysis, based on the preliminary findings of the second stage of analysis and literature review [23,24], our team operationalized definitions for the four categories of the GTE. These definitions were used by co-authors to integrate findings from the second stage of thematic analysis into a cohesive and pragmatic set of illustrative examples of equity-focused recommendations to navigate food insecurity in the context of early childhood programming (Fig 1).

## Results

### Characteristics of the interviewees

A total of 147 in-depth interviews with PCF teams, including municipal coordinators (n = 38), supervisors (n = 30), home visitors (n = 69), and municipal managers from sectors such as social protection, health, and education working directly with PCF teams (n = 10) were conducted across five Brazilian municipalities. The majority of professionals self-identified as females (89.1%), black or brown skin color (61.9%), older than 42 years (40.1%), having college education (49.0%), and 1–4 years of experience implementing programs (38.1%). A total of 93 in-depth interviews with families were conducted across five municipalities. The prevalence of food insecurity risk in households with children interviewed for this study ranged from 31.6% in Fortaleza to 73.3% in Campo Grande. Among caregivers interviewed, the majority self-identified as having black or brown skin color (83.0%), not employed at time of interview (70.2%), and being beneficiaries of the Brazilian conditional cash transfer program Bolsa Familia (69.9%) (S3–S4 Tables).



## Thematic analysis

 summarizes subthemes identified in the second phase of the thematic analysis. Themes were organized at the program, family, and system levels documenting the internal and external contexts in which PCF implementation operates.

**Theme 1. The internal context in which PCF operates.** *1.1. Program implementation lacks guidance on how to engage families navigating food insecurity.* All interviewees reported that the COVID-19 pandemic exacerbated existing economic and social vulnerabilities leading to **high prevalence of food insecurity risk among households participating in PCF**. Several families participating in PCF described challenges in accessing food due to limited income and high food prices. Specifically, caregivers shared that the uncertainty of having enough food was a major household stressor, which strained the relationships in the household, as described by a mother participating in the PCF:

> Here our financial conditions to buy everything to eat is difficult, it's a little tight, isn't it? (...) To be honest, I don't remember (...) when I last went to the store to buy lettuce or squash, or okra, a maxixe, I don't remember when I went (…) It's pretty bad [dealing with food insecurity]. Sometimes I get stressed. I take it out on my husband, but it's not his fault, is it? Thank God he works. But... things these days are so expensive. Everything is difficult (caregiver, Campo Grande).

Home visitors found it **challenging to engage caregivers struggling with food insecurity in early learning activities** such as play, games, and fun activities. They questioned whether teaching these activities should be a priority when people may be experiencing hunger. From their perspective, these activities designed to primarily strengthen the bond between caregiver-child to promote early learning are neither feasible nor appropriate in the context of many competing social needs. In this context, from the perspective of home visitors and families, the effectiveness of learning activities available from PCF was undermined by the food insecurity experience, as emphasized by a home visitor:

**Table 1. Themes and subthemes associated with food insecurity in households with children enrolled in the Brazilian *Criança Feliz* Program.**

| Themes | Level | Subthemes | Sub codes |
|---|---|---|---|
| Theme 1. The internal context in which PCF operates | Program | 1.1 Program implementation lacks guidance on how to engage with families navigating food insecurity | • High prevalence of food insecurity among PCF families |
| | | | • Challenges to engage food-insecure families in early learning activities |
| | | 1.2 Program curriculum lacks protocols to guide the implementation team in addressing food insecurity | • Lack of official guidance on how to navigate food insecurity |
| | | | • Lack of formal resolution for severe levels of food insecurity |
| Theme 2. The external context in which PCF operates | Family | 2.1 Family expectations concerning the program | • Expectations of stipends or food baskets tied to PCF participation |
| | | 2.2 Family involvement in food and income assistance programs | • Prioritization of PCF families to receive a one-time food stipend |
| | | | • Participation in the national conditional cash transfer program |
| | System | 2.3 Formal networks to enhance food security | • Lack of social assistance sector preparedness |
| | | | • Daycare can be a protective factor against severe food insecurity |
| | | 2.4 Informal networks to enhance food security | • Informal network among PCF home visitors and local/community resources |

Many [of the caregivers] do not see this positive side [in a child's early learning provided by the PCF]. It's very difficult to make a family that is often hungry understand the objective [of the PCF], what's gained in learning, [what is the point] of playing with the child (home visitor, São Paulo).

*1.2. Program curriculum lacks protocols to guide the implementation team in addressing food insecurity.* When asked about protocols to assist households at risk for food insecurity to access food, home visitors and supervisors reported the **absence of a government official guidance on how to navigate food insecurity** within PCF. The most immediate response from PCF teams to address food insecurity was finding ways of accessing food baskets to donate to families in need. However, PCF teams reported that the supply of food baskets was insufficient to meet the demand from families in need, as reported by a home visitor:

The difficulty we have at the moment is to meet people's demands because hunger is constant and food is a daily necessity. (…) We have a low supply [of basic food baskets] and a huge demand. There's never enough money (…). And that's what makes everyone anxious (…), but often our hands are tied, because we can't help all the families. (…) It's very complicated. (home visitor, Cuité).

Despite the lack of specific official guidance to manage screening and referral of families reporting having difficulties accessing food, home visitors used available resources to mitigate the issue. For example, they mentioned getting food baskets to households, which involves connecting families with official social protection services. The process of connecting with social protection services was described as bureaucratic, hierarchical, exhausting, and many times did not result in solutions for families, as highlighted by a home visitor:

The person needs a food basket (…) You have to make a request, and you go to the supervisor, and the supervisor goes to I don't know where and nothing happens. When you go to see the person, they haven't even received anything (home visitor, Campo Grande).

The lack of resources and guidance generated a lot of frustration among home visitors when working with food-insecure households. Indeed, some home visitors shared that **when no formal resolution for cases** was found through the official social protection services, they used their personal or collective financial resources to purchase food baskets for families, as stated by a home-visitor:

When you try to get a basket and you can't find it. Then, you have to do it yourself. But sometimes you also do not have the money to buy it, because nowadays everything is very expensive. Then I started asking [colleagues to chip in with money] to buy a food basket (…) (home visitor, Brasilia)

Although reporting informal resolutions, home visitors understood these were not long-term solutions for families. However, most of them shared that in critical situations like hunger, they could not move on with their day without providing any type of support to the families.

**Theme 2. The external context in which PCF operates.** *2.1 Family expectations concerning the program.* Families with young children assisted by the PCF reported the COVID-19 pandemic as a major event exacerbating economic strains (e.g., lack of money for food and other competing basic needs) and social vulnerabilities (e.g., unemployment, domestic violence, household stress). Coping strategies were also reported to manage strains and vulnerabilities. Specifically, families struggling with food were prone to ask for help from other people to get food, skip meals or not eat meals to save food for their children's next meal. When families were asked whether PCF could help them navigate food insecurity, caregivers expressed the **expectation that PCF would provide stipends or food baskets tied to program participation**, as illustrated by the following quote:

How difficult is that? Because we don't have any benefits [participating in PCF], and we eat what we have, [with the] help of other people, so we have to save as much as possible, especially for our children, we stop eating to give to our children (…) [The PCF] could help me get something. Especially with food stamps, right? With gas, everything helps… (mother, Fortaleza).

*2.2. Family involvement in food and income assistance programs.* It is important to acknowledge that the families' expectations were unrealistic since there is no financial assistance tied to PCF participation. However, supervisors and home visitors from one municipality shared a unique experience that proved two opposite points. On the positive side, policies to mitigate food insecurity can be rapidly enacted. On the negative side, rapidly enacted policies can pose a threat to program implementation, i.e., leading to confusion about the inclusion criteria of one-time benefits and the continued home visits offered by PCF During the pandemic, a specific municipal government decided to **prioritize families participating in PCF to receive a one-time food stipend** (R$250 [~US$50]). PCF teams were responsible for confirming families that were actively participating in program activities and the social assistance secretary was responsible for registering eligible families in the food stipend program. However, this one-time effort to mitigate food insecurity had limited coverage and left many families in need without access, which created confusion among families as to why some received the stipend and others did not, as highlighted in the quote from a home visitor:

As the PCF team, we confirmed the list of people active in the program, including information on those who have already received the food stipend program, [those who] are no longer receiving it, and [those who] will receive again (…) The process that takes the longest to put these people in the food stipend program is the interview with the [social assistance] worker. Thus, the only thing we [as a PCF home visitor] can do for those asking about the food stipend program is to request [the families] to call again [to schedule an interview with the social assistance worker] (home visitor, Brasilia).

In a way or another, PCF families often found themselves without adequate support when experiencing food insecurity. **Increasing family income through participation in the national conditional cash transfer program** (i.e., Bolsa Familia Program) emerged as a facilitator to reduce economic vulnerabilities, which in turn would directly affect food insecurity. Despite that, several home visitors expressed concerns about the COVID-19 pandemic-era government that changed some of the eligibility criteria of Bolsa Familia, reducing the number of eligible families. From the perspective of those interviewed, this was worrying because it could increase food insecurity among children. Specifically, families stated that the monthly cash transfer was used to buy food for their children, which in turn shielded them from food insecurity, as reported by a caregiver:

My husband is unemployed, we are surviving, getting hourly day work here and there; then he buys a little something, however, our child's food is specifically [bought with money] from Bolsa Familia. I just buy her food, and the rest of us make do (caregiver, Cuité).

*2.3 Formal networks to enhance food security.* PCF managers recognized challenges in promoting sustainable solutions to mitigate food insecurity due to **bureaucratic processes in accessing government safety net (i.e., social protection services) as well as limited non-government safety net (i.e., community resources)**. At the time of data collection, when families shared with home visitors, they are experiencing hunger, particularly not having food for the day; then, home visitors would notify supervisors, who in turn would notify PCF managers. Managers were able to refer these families to municipal social assistance centers. These municipal social assistance centers are responsible for assessing the needs of each family and enrolling them in as many social programs as they are eligible for such as Bolsa Familia, as seen in a home visitor's statement:



"So, we bring to municipal social assistance centers [the needs of each food insecurity family]. Then municipal social assistance centers within the social assistance sector make the move to call families and schedule a time for them to attend an in-person meeting at the municipal social assistance center. When families attend this meeting, they [families] receive what they need for example a food basket, rental assistance, and sometimes, even financial assistance with medication, when the family is very vulnerable, because many are. So, everything comes through the municipal social assistance centers" (home visitor, Cuité).

According to the families and home visitors interviewed, there was a huge waitlist with waiting times varying from one to seven months or more just to schedule an initial interview, which is required to access the services offered by the municipal social assistance centers. These waiting times were reported to be increased due to the high demand during COVID-19. The slow and bureaucratic process to get enrolled into the municipal social assistance centers resulted in families not having prompt access to social assistance programs to help them with economic vulnerabilities. The bureaucratic process was described by the interviewees as a lack of system preparedness during the pandemic. All interviewees agreed that the municipal social assistance sector had limited resolutions to address timely the complexity of food insecurity, as one manager noted:

"It's noteworthy. The problem of food insecurity exceeds what the social assistance sector can offer at all levels (…) both the municipal social assistance centers and our [social assistance] sector are not prepared and cannot address this complex issue of food insecurity" (PCF manager, São Paulo).

In this context, other community resources such as **daycare emerged as protective factors against hunger** among young children participating in the PCF. Many families stated that their children depended on the food provided by daycare centers. Thus, the shutdown of daycares during the COVID-19 pandemic meant that their children consumed less food. Therefore, the return of daycare operations was considered by all interviewees a positive measure to mitigate food insecurity among young children participating in the PCF, as highlighted by a caregiver:

"(…) now that school [daycare] is back on, children aren't so bad off. Because they have food at school. But it's lunch and dinner. Sometimes there's no breakfast, no snack. Understand?" (caregiver, Brasilia)

*2.4 Informal networks to enhance food security.* In search for additional resources to support food-insecure households, home visitors reported engaging informally with community organizations, including non-profit neighborhood organizations, churches or religious places, and local grocery shop owners. Home visitors shared that **building this informal/ non-governmental network** was strategic particularly during the COVID-19 pandemic as they used it to obtain donations of food baskets to families in urgent need of food, as highlighted in a home visitor quote:

"(…) We do it like this: in this case, I seek out [to build a relationship with a] neighborhood association (…) When I notice that a family is in a very vulnerable situation, then I ask [to the neighborhood association] if they have a food basket to give [to that family in need], understand? (home visitor, Fortaleza)

Home visitors acknowledged that many times they were not able to obtain food baskets through these informal networks, which also depend on outside donations. On one hand, building this informal network was valuable for their work. On the other hand, the informality of these relationships made it hard to have it as a sustainable option for food-insecure families.

## Data integration to inform equity-focused recommendations

Informed by the data analysis, definitions for the four GTE categories were operationalized to integrate findings and develop equity-focused recommendations to address food insecurity in early childhood programs. Program internal

context was defined as potential interventions in program implementation and curriculum and included two dimensions: (1) reduce deterrents to program implementation (i.e., strategies to screen and manage food insecurity), and (2) increase healthy food (i.e., strategies to improve participant access to healthy food). Program external context was defined as potential interventions to support individual participants and community resources and capacity and include two dimensions: (3) improve social and economic resources (i.e., PCF strategies to support alleviate participants' food insecurity in partnership with other programs), and (4) build community capacity (i.e., strategic choice of community partnerships to alleviate food insecurity) (Fig 2).

Data integration based on adapted definitions for GTE categories supported the development of equity-focused recommendations related to the program internal context included strategies to **reduce deterrents** to program implementation (e.g., staff training, development of protocols to screen, refer, and follow up with families at risk for food insecurity, protocols for immediate access to food in case of hunger), and strategies to **increase access to healthy food** by tailoring program curriculum (e.g., address fear of discrimination/judgment for experiencing food insecurity, which may lead to reduced participation in assistance programs, and share with families affordable healthy, budget friendly options) (Table 2).

Recommendations related to the program external context outline strategies to **improve social and economic resources for families** (e.g., support families' enrolment in social and/or food assistance programs, share resources and programs with families) and **build community capacity** to enhance the safety net system to promote food security (e.g., develop protocols and leverage community assets to address food insecurity and other family vulnerabilities, and build partnerships) (Table 2).

## Discussion

This study documented barriers and opportunities to equitably address food insecurity among families with young children enrolled in one of the largest early childhood programs worldwide. Within the internal context of PCF, the absence of implementation protocols to screen, refer, and follow up challenged the engagement of food-insecure households in early learning activities. Within the external context of the PCF, family-level factors included expectations of receiving stipends or easy access to food baskets, and system-level factors such as timeliness, bureaucracy, and lack of preparedness of government and non-government organizations to support food-insecure households in face of a global pandemic. These factors were found to impact early learning and responsive care environments possibly due to food and nutrition restrictions [37], parental depression, stress, and anxiety [38], which in turn, may increase inequities in early childhood development and perpetuate the cycle of poverty [7–9]. The documentation of these barriers underscored the critical need for equity-focused recommendations to address food insecurity within early childhood early childhood home visiting programs. This study findings informed the development of a not exhaustive list of equity-focused recommendations grounded in the GTE that can enhance early childhood program implementation as well as individual and community resources and capacity (see Table 2).

Within the internal context of the PCF, this study identified high prevalence of households with young children at risk for food insecurity ranging from 31.6% to 73.3%. These numbers corroborate with global trends for maternal-child populations during and after the COVID-19 pandemic [39–41]. Understanding food insecurity as a transient phenomenon with most households moving in and out of food security as family circumstances change [42] is important to propose recommendations that allow time for vulnerable families with children to recover from severe disruptions. This is particularly concerning because food insecurity adversely impacts child health and development [7–9]. Therefore, specifying recommendations to enhance the implementation capacity of early childhood programs to support families in transitioning out of food insecurity and sustaining food security status is crucial.

Protocols to regularly screen, refer, and follow up households at risk for food insecurity can reduce deterrents to program implementation. Incorporating food insecurity screening as a routine during early childhood home visits can be a low-burden strategy to establish continuity of care. Adoption of brief screening tools such as the validated two-item food

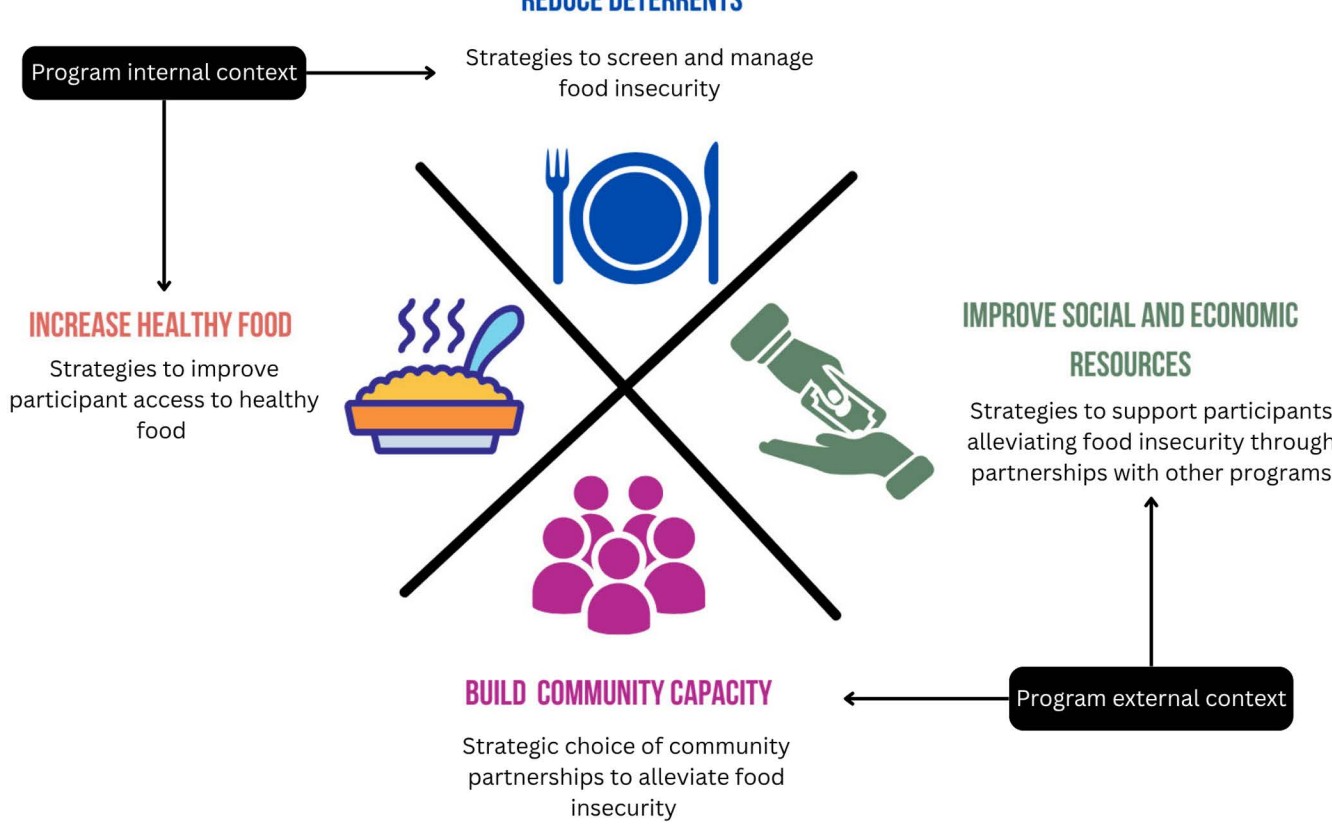

**Fig 2. Adapted definitions for Getting to Equity categories to support the development of equity-focused recommendations for addressing food insecurity in early childhood programs.**

insecurity screen (TRIA) [32]could be used to identify households at risk for food insecurity. In case of positive screens, early childhood home visiting teams would rely on multidisciplinary services personnel to classify the severity level of food insecurity through a diagnosis scale such as the 14-item Brazilian Food Insecurity Scale (EBIA) [43] which is based on the Household Food Security Survey Module (HFSSM) [44]. The identification of food insecurity severity level is critical to establish a targeted action plan and ensure equitable resource allocation. Therefore, the development of early childhood home visiting program protocols would benefit from the inclusion of decision-making trees with (1) incorporation of a validated screening tool, (2) classification of food insecurity severity level, (3) determination of a referral pathways for each severity level of food insecurity, and (4) inclusion of messages/actions based on the governance of the early childhood program.

Tailoring short-, medium-, and long-term messages/actions to the PCF curriculum is essential for improving the impact on early childhood outcomes. While addressing food insecurity, there is a specific opportunity to enhance the early childhood curriculum by integrating evidence-informed messages on nutrition literacy, including food access and healthy eating. Evidence from programs implemented in middle- and low-income countries demonstrated that integrating nutrition messages into early childhood programs is effective [45,46]. The integration of messages requires an understanding of how nutritional contexts and caregiver characteristics such as well-being, knowledge, and behavior may influence early child nutrition outcome [45]. Accordingly, as part of the implementation plan, early childhood home visiting teams should be trained in the integrated curriculum and protocols to address food insecurity. In addition, they should be educated

**Table 2. Equity-focused strategies (not exhaustive) for addressing food insecurity in early childhood programs.**

| Getting to Equity (GTE) dimensions | | Illustrative Examples of Programming Recommendations for Early Childhood Programs |
|---|---|---|
| Program Internal Context: potential interventions in program implementation and curriculum | Reduce Deterrents (program implementation) | • Staff trained on the impacts of food insecurity on mental health, infant feeding, and early childhood development; |
| | | • Create protocols to screen, refer, and follow up with families at risk for food insecurity, including why, how, and what; |
| | | • Create protocols for immediate access to food in case of moderate and severe levels of food insecurity; |
| | Increase access to healthy food (program curriculum) | • Address fear of discrimination/judgement due to experience food insecurity; |
| | | • Create navigation resources to educate families about the available assistance programs and options to help them manage their income. |
| | | • Provide resources on healthy, culturally tailored, budget-friendly food options to families who request assistance. |
| Program External Context; potential interventions to support individuals and community resources and capacity | Improve social and economic resources (families) | • Support families to seek ways of increasing household minimum wage through enrolment in social or food assistance programs; |
| | | • Assist caregivers with child and family needs by sharing community resources and other programs; |
| | Build on community capacity (system) | • Create protocols to facilitate warm handoff to services aiming at reducing family's social and economic vulnerabilities; |
| | | • Document and leverage existing multi sectoral community assets to support food-insecure families such as daycares; |
| | | • Build formal strategic partnerships to strengthen to facilitate access to healthy food for families in the program. |

about their role in supporting families navigating food insecurity. Training and supportive supervision are important implementation strategies to reduce stress and retain the workforce of early childhood programs [47]. Indeed, training early childhood home visiting teams to recognize food deprivation is critical to reducing social disparities [48].

Within the external context of the PCF, family-level factors included expectations of receiving stipends or facilitated access to food baskets tied to PCF participation. Thus, one recommendation would be to better advertisement of what the PCF entails to correct expectations. While these expectations were not met by the PCF in Brazil, our findings documented the efforts of PCF teams to enroll families in Bolsa Familia as one strategy to reduce the risk for food insecurity. There is strong evidence that combining benefits such as cash transfers or food benefits (e.g., food basket, food stamps) to vulnerable families enrolled in early childhood and parenting programs is beneficial to addressing food insecurity as well as to reducing the inequality gap [49,50]. Several institutional ways for combining early childhood program and cash transfers programs have been documented globally, including *integrated* (i.e., the parenting intervention is managed by Bolsa Familia, *convergent* (i.e., different agencies explicitly combine efforts to bring the separate cash transfer and parenting programs to the same populations), *alignment* (i.e., the cash transfer and the parenting programs do not explicitly coordinate with one another but deliver interventions to similar if not the same populations), and *piggybacking* (i.e., the cash transfer is delivered through a separate established platform) [49]. This study findings indicate that in Brazil there is an alignment between early childhood development and conditional cash transfer programs as both programs target similar populations. However, a convergent approach where both programs are integrated and mutually supportive could address risk for food insecurity, additional social needs, and enhance the impact on early childhood outcomes, as demonstrated in other middle-income countries with similar governance systems to Brazil [49].

This study findings documented system level barriers such as the timeliness and bureaucracy related to the scarce government and non-governmental social safety net to support food-insecure households with children. There is strong evidence that both government and non-government safety nets can play a key role in protecting against food insecurity while families cope with the effect of additional social needs. Government safety nets may include participation in government benefits such as food [51] assistance or cash transfer programs as discussed above. Additionally, results showed participation in school/daycare programs was one important mechanism in the government safety net to support food-insecure households with children. Evidence shows that universal and free school meals contribute to children's nutrition and households' food security [52]. Brazil's early education and care system provides universal free-of-cost daycare for about 50% of the population [53]. However, due to the COVID-19 pandemic, daycares shut down. Without the food support [54], several PCF families struggled to provide food for their children. As a result, PCF teams experienced pressure to fill the gap left by the government safety nets. Thus, this study documented strategies to build a non-governmental safety net to support food insecurity households with children. Non-governmental partnerships with community-based non-profit organizations can be a successful strategy for immediate relief of hunger, and it has been a mechanism for civil society to engage in the fight to end hunger in Brazil since the 90's [55]; however, food nonprofits need more government support to be sustainable [56]. This will require national and local political efforts to establish and sustain continuity of care to navigate food insecurity [57]. Recent cross-sectoral collaborations involving public and private institutions, such as the Pact Against Hunger are good examples of strategic and sustainable partnerships aiming to eradicate hunger in Brazil [58]. Formalizing strategic partnerships between early childhood development programs and multisectoral safety nets acknowledges that addressing children's food and nutrition needs is critical to achieving equity in early childhood development.

This study's limitation includes a small sample of Brazilian municipalities. Specifically, the selection of four state capitals and one small municipality left medium-sized municipalities unrepresented. Despite that, implementation challenges to address food insecurity were similar in the included municipalities, suggesting that findings may have captured program, families, and system barriers that are likely to be encountered across municipalities independent of population size. Additionally, the inclusion of interviews with PCF teams and families across targeted groups (i.e., pregnant women, children younger than three years) allowed for data saturation on challenges to address food insecurity at the program, families, and system levels [33]. One possible bias during data collection is social desirability. Because food insecurity might be a sensitive topic for some participants, they may have altered their responses during the interviews to present themselves in a favorable light, fearing negative judgment or disapproval. To address this, our team of interviewers fostered a secure environment for participants to share their experiences without judgment. Many shared very challenging situations related to food insecurity and hunger. This suggests our research team provided a safe interview environment, which may have reduced the risk of social desirability bias. As a result, these findings and recommendations reflect themes that emerged from experiences of interviewees involved in implementing the PCF in urban and rural contexts across different Brazilian regions.

This study's strengths include the use of rapid analysis which demonstrated solid and replicable results compared to traditional in-depth qualitative coding analysis [34,59]. Thus, this approach is not only innovative but also provides timely knowledge translation into public health recommendations. The deductive application of the GTE framework to integrate findings into practical programming recommendations is also a strength. On the other hand, we acknowledge that the non-exhaustive list of recommendations outlined in this study represents and it is limited to the perspective of the co-authors, who are researchers working with evaluation of implementation programs related to food insecurity and/or early childhood development in Brazil for several years. Therefore, the non-exhaustive list of recommendations is informed by this study's findings through the lens of their lived experiences and aligned with the most current literature on these topics. GTE has been extensively used to analyze dimensions of equity in obesity prevention [23] and access to healthy food [60] programs and in this study was instrumental in guiding specific and relevant programming recommendations to address food insecurity within early childhood programming to achieve equity in early childhood development.

                                                                        

In conclusion, this study documented factors at the program, families, and system levels to address food insecurity among households with young children enrolled in the Brazilian early childhood development home visiting program. Grounded in the GTE framework, this study integrated opportunities identified in actionable equity-focused programming recommendations that may enhance effectiveness of one of the largest early childhood development programs worldwide (see Table 2). During this period of reorienting PCF implementation, it will be crucial to consider the following key actions: (1) incorporate universal screening for risk of food insecurity along with protocols to empower PCF teams to navigate food insecurity, (2) develop messages to enhance the PCF curriculum on child food and nutritional needs and its impact on early childhood, (3) enhance capacity of government safety net programs, including Bolsa Familia, food benefit programs (e.g., food basket, food stamps), and early education and care through a convergent integrated approach to address the food insecurity, (4) establish a local non-governmental safety net by formalizing multisectoral partnerships to facilitate continuity of care for food insecurity households and promote equity in early childhood development.

## Supporting information

**S1 Table. *Criança Feliz* Program national guidelines following the Template for Intervention Description (TIDieR).**
(DOCX)

**S2 Table. Consolidated criteria for reporting qualitative research checklist (COREQ).**
(DOCX)

**S3 Table. Socio-demographic characteristics of the professionals interviewed in the *Criança Feliz* Program (PCF), 2021–2022.**
(DOCX)

**S4 Table. Socio-demographic characteristics of the families interviewed in the *Criança Feliz* Program (PCF), 2021–2022.**
(DOCX)

**S5 Table. Selected questions of the Interview Guide related to Food insecurity, vulnerability, and equity, according to the RE-AIM Framework dimensions (Reach, Effectiveness, Adoption, Implementation, Maintenance).**
(DOCX)

## Acknowledgments

We would like to thank Dr. Sonia Isoyama Venancio, and research assistants Lidia Godoi and Laura Dal'Ava dos Santos for their assistance and support during this study.

## Author contributions

**Conceptualization:** Gabriela Buccini.

**Formal analysis:** Poliana de Araújo Palmeira, Muriel Bauermann Gubert.

**Funding acquisition:** Gabriela Buccini.

**Methodology:** Gabriela Buccini.

**Supervision:** Gabriela Buccini.

**Writing – original draft:** Gabriela Buccini.

**Writing – review & editing:** Gabriela Buccini, Poliana de Araújo Palmeira, Ana Poblacion, Muriel Bauermann Gubert.



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
