## [Decision Letter · Decision Letter 0]

PONE-D-24-37907Addressing Food Insecurity in Early Childhood Programs through a Health Equity Lens: A Qualitative Case Study of Brazil’s Criança Feliz Program.PLOS ONE

Dear Dr. Buccini,

Thank you for submitting your manuscript to PLOS ONE. After careful consideration, we feel that it has merit but does not fully meet PLOS ONE’s publication criteria as it currently stands. Therefore, we invite you to submit a revised version of the manuscript that addresses the points raised during the review process.

We look forward to receiving your revised manuscript.

Kind regards,

Hamufare Dumisani Mugauri, Ph.D. Medicine and Health Sciences

Academic Editor

PLOS ONE

Journal Requirements:

Journal Requirements:

Reviewers' comments:

Reviewer's Responses to Questions

**Comments to the Author**

1. Is the manuscript technically sound, and do the data support the conclusions?

Reviewer #1: Yes

Reviewer #2: Partly

Reviewer #3: Partly

Reviewer #4: Yes

2. Has the statistical analysis been performed appropriately and rigorously? 

Reviewer #1: N/A

Reviewer #2: N/A

Reviewer #3: N/A

Reviewer #4: Yes

3. Have the authors made all data underlying the findings in their manuscript fully available?

Reviewer #1: No

Reviewer #2: No

Reviewer #3: No

Reviewer #4: No

4. Is the manuscript presented in an intelligible fashion and written in standard English?

Reviewer #1: Yes

Reviewer #2: Yes

Reviewer #3: Yes

Reviewer #4: No

5. Review Comments to the Author

Reviewer #1: Dear Authors,

Congratulations on an impressive and innovative article! The manuscript is well-written and offers valuable insights highly relevant to the creation and implementation of policies. I have provided a few minor suggestions, but overall, your work is thorough and thoughtfully crafted.

Thank you for your contribution to this important field.

Best regards!

Reviewer #2: I think this is a brilliant study, and provides some vital insight and guidance as to how these programs can be further improved to protect against food insecurity. I also thought it was very clear and well-written.

However, I would like to suggest some improvements to the link between the results and the discussion, including the Getting to Equity framework. Firstly, whilst this framework is a brilliant way to organise recommendations, I think in places it feels a little shoehorned in. The Nutrition and Food Security framework may also be of benefit to reference to, especially in the introduction when discussing the bigger picture of food insecurity. This would also better link consequences such as stress (mentioned on page 4), which is currently referred to as a mechanism.

This is also the case in the discussion - see comments below.

Please clarify the logistics of the PCF, is this state run or run by an NGO or CSO?

The different demographics and locations that have been taken into account to make the sample as generalisable as possible - big strength of the study.

Please provide brief details of the TRIA screening tool - what dimensions of FI do they look at?

On page 10, line 199 - it could be argued that it is underlying equities that cause FI, and therefore FI is not the root cause but a metric to understand who is most affected by the root issues. The equities that the GTE framework is addressing contribute to FI as a consequential issue.

Line 202/203 - were the GTE recommendations adapted before or after the results were considered? If it was after, should this be in Results as a finding of the study to inform the adaptation of the framework. If it was adapted through literature searches then just clarify that this is how you came to it.

Line 204: Is this deterrents to participating in the programs? As the initial framework is for obesity, this would usually refer to reducing deterrents that cause obesity. Here,you are reflecting it for FI, but I understand you are not reducing deterrents to be food insecure, so should it rather be reducing barriers to the program and engaging with it etc.

Table 1: What does high frequency and levels mean? Is there a high prevalence, and within this a lot of people are severely food insecure?

Line 296: Tie some of the results back to the objectives more clearly, looking at the effects on the children.

Line 315: Please clarify here who's governance this affects or the relevance.

Line 411: How would this help to address fear of discrimination? Discrimination of or due to what?

Table 2 & discussion: If affordability is the biggest issue, should it be increasing access to affordable healthy, budget-friendly options. Some families may know what to do but the finances seems to be the biggest barrier here. As the quotes suggest, engaging with the children on games etc didn't seem to make too much of an impact when families were hungry. I assume the same will be the case if households are starving, they will not want to sit through talks about food. It would be beneficial if the GTE framework better reflected this - as there feels to be a sort of disconnect between the findings demonstrating that affordability and a lack of financial help or food baskets is the issue and the recommendations?

Also impressive to see a return of data to the community. Clear impact in discussion demonstrated.

Also, the data is only available on request and is not publicly available.

ECD abbreviation not stated before first use.

Line 477: Could a recommendation be better advertisement of what the program entails to correct expectations?

Reviewer #3: Overall

- This is an important and timely topic. The manuscript provides a comprehensive descriptive analysis of food insecurity in Brazil. The manuscript is well-written and organized.

- The use of multiple uncommon acronyms throughout the manuscript can be confusing for readers. Please reduce the number of acronyms and limit them to conventionally used ones. Avoid using the acronym “FI.”

Introduction

- The introduction defines food security as a lack of access to food but does not address its multidimensional aspects (e.g., availability, utilization, stability). Please include these dimensions https://doi.org/10.4060/cc3017en

- There appears to be a typo on line 64.

- Lines 60-64 mention that addressing food insecurity is a complex problem but do not explain why. Please elaborate on the complexity beyond just its prevalence.

- Define key terms when first introduced, such as “social determinants of health” on line 81.

Methods

- Are the five municipalities representative of Brazil?

- Lines 123-126: Please clarify your sampling approach. What specific characteristics were you targeting? Is “e.g.” used correctly, or should it be “i.e.”?

- Confirm if all interviews were conducted virtually (line 127). Did any include video?

- On line 145, clarify the term “loosened.” Provide an overview of COVID-19 restrictions prior to June 2021 and any changes during data collection.

- Confirm whether participants provided consent before interviews.

- Explain why in-depth interviews were chosen as the best method for answering your research question. Provide justification for this approach.

- Consider including the interview guide or key questions relevant to this study for transparency.

- Clarify lines 182-184 regarding if the findings were presented as a form of member checking and how this feedback informed further analysis.

- Specify how food security was defined in the methods. What tool or criteria were used?

Results

- Please include more descriptives in the response to the quotes. Important information for caregivers that provides necessary context but does not reveal any identifiable information for the caregivers is household size and food security status. Please also consider what important and relevant information is for the other participant sources and include these in the results section.

- The current organization by levels (e.g., family, system) makes it difficult to clearly see how the results address the research question (barriers and facilitators). Please reorganize or explicitly align the results with the research question to improve clarity and linkage to the discussion.

Discussion

- The linkage on the results to ECD is not strong. Avoid making broad claims unsupported by the data.

- "Public health" should not be capitalized on line 531.

- Discuss potential biases that may have influenced participant responses and include them in the limitations section.

Reviewer #4: All: The article needs to be reviewed for spelling, grammar and academic language. There are quite a few language and punctuation issues throughout, and issues with formatting of the references.

Abstract: I think that 'home visiting' needs explaining earlier on - as it assumes knowledge.

Keyword: I think the article would benefit from Keywords.

Introduction: I think it would be welcome to provide some illustrative examples of the ten recommendations in the introduction.

There are some unqualified statements such as 'as no children should go hungry' which need more academic rigour attached to them.

Methods: I think the theories behind the methods need to be summarised up front.

I think the methods section would benefit from research questions, and subsequently how the methods used were chosen to answer the research questions.

The method structure should be reviewed for clarity and ensuring the right information is in the right headings. Data management talked about when and how visits were conducted, which wasn't related to data management for example.

Line 249 - what are Dyads?

Results: The results section would benefit from having better signposting and sub-sectioning of the themes, as it was at times hard to follow.

6. PLOS authors have the option to publish the peer review history of their article (what does this mean? ). If published, this will include your full peer review and any attached files.

**Do you want your identity to be public for this peer review?** For information about this choice, including consent withdrawal, please see our Privacy Policy .

Reviewer #1: No

Reviewer #2: **Yes: ** Jessica Boxall

Reviewer #3: No

Reviewer #4: No

---

## [Author Response · Author response to Decision Letter 1]

2 Jun 2025

Dear Editor,

Thank you for the opportunity to revise our manuscript. Below are the response point-by-point to journal requirements and reviewer’s feedback. We hope our revised version meets your expectations.

Journal Requirements:

Response: Done.

2. We note that you have indicated that there are restrictions to data sharing for this study. Please update your Data Availability statement in the submission form accordingly.

Response: A de-identified data set is not possible to provide due to ethical and legal restrictions. These sharing restrictions are imposed by the UNLV Institutional Board Review (IRB). The authors declare that a de-identified data set from this study are available upon request directly to Dr. Buccini (gabriela.buccini@unlv.edu) and/or to the UNLV IRB (irb@unlv.edu).

Reviewer #1:

Dear Authors, Congratulations on an impressive and innovative article! The manuscript is well-written and offers valuable insights highly relevant to the creation and implementation of policies. I have provided a few minor suggestions, but overall, your work is thorough and thoughtfully crafted. Thank you for your contribution to this important field. Best regards!

Response: Thanks for the positive feedback.

A#1: L. 51. How is FI related to stress?

A#1: L. 23. How is FI related to stress?

Response: We expanded the background section to organize the pathways causing food insecurity and clarify how stress is related to food insecurity (lines 67-76).

A#1: L. 63. This part could be removed, and the phrase could end at FI

Response: Removed in the new version of the background.

A#1: L. 64. I believe this is a typo

Response: Corrected.

A#1: L. 67. This topic is a bit confusing

Response: Removed in the new version of the background.

A#1: L. 74. Is PCF based on NCF?

Response: Yes, it is. This has been added in the new version of the background.

A#1: L. 170. This part is a bit confusing

Response: Reworded to clarify.

A#1: L. 126. Receiving?

Response: We kept the wording “beneficiaries” based on the following: in general, "beneficiary" and "recipient" are often used interchangeably to describe someone who receives something. However, "beneficiary" often implies a more formal or legally defined receiving of benefits, while "recipient" is broader and can apply to any kind of receiving.

A#1: L. 162. Relevant to this study

Response: Changed.

A#1: L. 175. I believe this is also a typo

Response: We could not find the typo.

A#1: L. 178. was

Response: Corrected.

A#1: L. 180. In order to?

Response: Added.

A#1: L. 246. food insecurity?

Response: Corrected.

A#1: L. 366. "... just to schedule an initial interview, which is required to access services offered by the municipal social assistance centers."

Response: Corrected.

Reviewer #2:

I think this is a brilliant study, and provides some vital insight and guidance as to how these programs can be further improved to protect against food insecurity. I also thought it was very clear and well-written.

Response: Thanks for the positive feedback.

However, I would like to suggest some improvements to the link between the results and the discussion, including the Getting to Equity framework. Firstly, whilst this framework is a brilliant way to organise recommendations, I think in places it feels a little shoehorned in. The Nutrition and Food Security framework may also be of benefit to reference to, especially in the introduction when discussing the bigger picture of food insecurity. This would also better link consequences such as stress (mentioned on page 4), which is currently referred to as a mechanism. This is also the case in the discussion - see comments below.

Response: We reorganized the background and information concerning the Nutrition and Food Security framework was included as suggested (lines 53-60).

L. 47. There is a wider network of factors that contribute to food insecurity. See the Nutrition and Food Security Framework, maybe useful to cite here and provide a bigger picture.

Response: This has been added in the new version of the background.

Please clarify the logistics of the PCF, is this state run or run by an NGO or CSO?

Response: To clarify this question, we expanded information about the PCF in the revised introduction as follows: The PCF uses a multilevel implementation strategy across the three administrative levels of the Brazilian government. The national level is responsible for funding, articulating the multisectoral approach, and coordinating the implementation by developing protocols, training, and monitoring strategies. The state level provides technical support to municipal teams. At the municipal level, teams are responsible for home visits and complementary multisectoral actions (Buccini et al., 2021,2024).

A#JB: L. 54. Is this a mechanism or rather a consequence of food insecurity causing stress on the family? Please clarify the relationship here.

Response: We expanded the explanation of the pathways between food insecurity and early childhood development. Specifically, the socio emotional path when stress is activated can act as both consequence or mechanism exacerbating stress leading to higher risk of food insecurity.

A#JB: L. 54. Is this a typo for stresses?

Response: Corrected.

A#JB: L. 64. Seems to be a rogue abbreviation!

Response: Removed in the new version of the introduction.

A#JB: L. 71. Is this state run?

Response: add more on the program

The different demographics and locations that have been taken into account to make the sample as generalisable as possible - big strength of the study.

Response: Thank you.

Please provide brief details of the TRIA screening tool - what dimensions of FI do they look at?

Response: Both interview guides had a section on sociodemographic characteristics of participants, which included the validated two-item screening tool (TRIA) to identify Brazilian households at risk for food insecurity (sensitivity of 79.31%, specificity of 92.95%) [31]. Specifically, TRIA uses questions 2 and 4 of the Brazilian Food Insecurity Scale (EBIA): question number 2 in the EBIA (“Nos últimos três meses a comida acabou antes que você tivesse mais dinheiro para comprar mais?”) corresponds to HFSSM question Q2: “In the past 12 months, the food that (I/we) bought just didn’t last, and (I/we) didn’t have money to get more?”); and question number 4 in the EBIA (“Nos últimos três meses, você teve que se arranjar com apenas alguns alimentos para alimentar os moradores com menos de 18 anos, porque o dinheiro acabou?”) corresponds to HFSSM question Q4: “In the past 12 months, did (you or other adults in your household) ever cut the size of your meals or skip meals because there wasn’t enough money for food?” [31].When a participant caregiver responded affirmatively to both items of the instrument, their household was considered at risk for food insecurity [31].

L. 123. Lots has been considered to make the sample as generalisable as possible. Impressive.

Response: Thank you

L. 166. Please provide brief details of these items and what dimensions of FI they look at.

Response: Added

On page 10, line 199 - it could be argued that it is underlying equities that cause FI, and therefore FI is not the root cause but a metric to understand who is most affected by the root issues. The equities that the GTE framework is addressing contribute to FI as a consequential issue.

Response: Thanks for this comment. We expanded the paragraph to include this important consideration as follows: Therefore, in our study, food insecurity is not the root cause of inequities rather is a metric used to understand who was most affected by the root causes (e.g. poverty). The equities that the GTE framework is addressing contribute to food security as a consequential issue.

Line 202/203 - were the GTE recommendations adapted before or after the results were considered? If it was after, should this be in Results as a finding of the study to inform the adaptation of the framework. If it was adapted through literature searches then just clarify that this is how you came to it.

Response: Thanks for this comment. We clarified that the GTE definitions were adapted informed by our findings; then, after adapting the GTE definitions, recommendations were developed. To make it clear in the manuscript, as suggested, we moved the definitions to the Results section.

#2: Line 204: Is this deterrents to participating in the programs? As the initial framework is for obesity, this would usually refer to reducing deterrents that cause obesity. Here, you are reflecting it for FI, but I understand you are not reducing deterrents to be food insecure, so should it rather be reducing barriers to the program and engaging with it etc.

Response: Thanks for this comment. We clarified that deterrents are related to program implementation, i.e., staff training and creation of protocols to help participants to navigate food insecurity.

Table 1: What does high frequency and levels mean? Is there a high prevalence, and within this a lot of people are severely food insecure?

Response: We edited to clarify “high prevalence” of food insecurity refers to participating families. Levels of food insecurity were not measured in our study nor by the PCF, so this wording was removed.

Line 296: Tie some of the results back to the objectives more clearly, looking at the effects on the children.

Response: As suggested by reviewer#3, we revised the objectives and now the results are tied to the objectives.

Line 315: Please clarify here who's governance this affects or the relevance.

Response: Added.

Line 411: How would this help to address fear of discrimination? Discrimination of or due to what?

Response: Discrimination due to experience food insecurity. We included the following statement: Fear of discrimination is a significant barrier for individuals experiencing food insecurity, leading to reduced participation in assistance programs and contributing to feelings of shame and stigma.

Table 2 & discussion: If affordability is the biggest issue, should it be increasing access to affordable healthy, budget-friendly options. Some families may know what to do but the finances seems to be the biggest barrier here. As the quotes suggest, engaging with the children on games etc didn't seem to make too much of an impact when families were hungry. I assume the same will be the case if households are starving, they will not want to sit through talks about food. It would be beneficial if the GTE framework better reflected this - as there feels to be a sort of disconnect between the findings demonstrating that affordability and a lack of financial help or food baskets is the issue and the recommendations?

Response: We agree with the assumption that families experiencing hunger may not benefit educational resources on healthy food, thus, as suggested, we reworded it in Table 2 as follows:

• Create navigation resources to educate families about the available assistance programs and options to help them manage their income.

• Provide resources on healthy, culturally tailored, budget-friendly food options to families who request assistance.

Also impressive to see a return of data to the community. Clear impact in discussion demonstrated. Also, the data is only available on request and is not publicly available.

Response: Thank you. Although data is available upon request, all municipalities received a detailed report about the findings in their municipality. Additionally, a full report comparing the five municipalities was made available to the Ministry of Citizenship and all municipalities implementing PCF.

ECD abbreviation not stated before first use.

Response: Thanks for catching it. This acronym has been removed in the effort to address a comment from reviewer#3.

Line 477: Could a recommendation be better advertisement of what the program entails to correct expectations?

Response: Added

Reviewer #3: Overall

- This is an important and timely topic. The manuscript provides a comprehensive descriptive analysis of food insecurity in Brazil. The manuscript is well-written and organized.

- The use of multiple uncommon acronyms throughout the manuscript can be confusing for readers. Please reduce the number of acronyms and limit them to conventionally used ones. Avoid using the acronym “FI.”

Response: Thanks for the positive feedback. We reduced the number of acronyms as suggested.

Introduction

- The introduction defines food security as a lack of access to food but does not address its multidimensional aspects (e.g., availability, utilization, stability). Please include these dimensions https://doi.org/10.4060/cc3017en

Response: This has been added to the background.

- There appears to be a typo on line 64.

Response: Corrected.

- Lines 60-64 mention that addressing food insecurity is a complex problem but do not explain why. Please elaborate on the complexity beyond just its prevalence.

Response: We reframed the background as suggested.

- Define key terms when first introduced, such as “social determinants of health” on line 81.

Response: We excluded the wording.

Methods

- Are the five municipalities representative of Brazil?

Response: As mention in the methods, municipalities were chosen based on different characteristics to represent different realities of Brazil. These characteristics included: (1) population size, (2) implementation length, (3) implementation model, and (4) geographical region. As mentioned in the discussion, the small sample size of municipalities is a limitation. Additionally, the large number of interviews allowed an in-depth analysis of implementation barriers, which is the main goal of a case-study design.

- Lines 123-126: Please clarify your sampling approach. What specific characteristics were you targeting? Is “e.g.” used correctly, or should it be “i.e.”?

Response: We added that a purposive sampling approach was used. The “e.g.” was replaced by “i.e.”.

- Confirm if all interviews were conducted virtually (line 127). Did any include video?

Response: In-depth interviews were conducted virtually (with or without video according to the participant preference) or via phone.

- On line 145, clarify the term “loosened.” Provide an overview of COVID-19 restrictions prior to June 2021 and any changes during data collection.

Response: Data collection occurred post-COVID-19, thus no adaptation or changes happened during data collection. To simplify and not confuse the reader this sentence has been removed.

- Confirm whether participants provided consent before interviews.

Response: Confirmed. This was mentioned in the first statement of the methods section.

- Explain why in-depth interviews were chosen as the best method for answering your research question. Provide justification for this approach.

Response: Justification added in the study design in the methods section.

- Consider including the interview guide or key questions relevant to this study for transparency.

Response: Interview guide was already included in the original submission as Supplementary Material 5.

- Clarify lines 182-184 regarding if the findings were presented as a form of member checking and how this feedback informed further analysis.

Response: We edited to clarify how this iteration informed the analysis presented in this manuscript. The preliminary findings of the first stage of thematic analysis were presented to each municipality for feedback, fostering reflection within the research team. Insights from this interaction between implementers and research team generated research questions related to food insecurity and guided the second stage of thematic analysis as well as the integration of data included in this manuscript.

- Specify how food security was defined in the methods

---

## [Editor Report · Decision Letter 1]

PONE-D-24-37907R1Addressing Food Insecurity in Early Childhood Programs through a Health Equity Lens: A Qualitative Case Study of Brazil’s Criança Feliz Program.PLOS ONE

Dear Dr. Buccini,

Thank you for submitting your manuscript to PLOS ONE. After careful consideration, we feel that it has merit but does not fully meet PLOS ONE’s publication criteria as it currently stands. Therefore, we invite you to submit a revised version of the manuscript that addresses the points raised during the review process. May you check the following before a final decision on your manuscript:

PLOS ONE considers qualitative and mixed-methods studies for publication. We recommend that authors use the COREQ checklist, or other relevant checklists listed by the Equator Network, such as the SRQR, to ensure complete reporting (http://journals.plos.org/plosone/s/submission-guidelines#loc-qualitative-research). In general, we would expect qualitative studies to include the following: 1) defined objectives or research questions; 2) description of the sampling strategy, including rationale for the recruitment method, participant inclusion/exclusion criteria and the number of participants recruited; 3) detailed reporting of the data collection procedures; 4) data analysis procedures described in sufficient detail to enable replication; 5) a discussion of potential sources of bias; and 6) a discussion of limitations. In your role as Academic Editor, we appreciate your consideration of whether the manuscript meets reporting standards in the field, in addition to the journal’s other publication criteria (https://journals.plos.org/plosone/s/criteria-for-publication).Please contact plosone@plos.org with any questions or concerns.

Please submit your revised manuscript by Aug 25 2025 11:59PM. If you will need more time than this to complete your revisions, please reply to this message or contact the journal office at plosone@plos.org . Please include the following items when submitting your revised manuscript:

We look forward to receiving your revised manuscript.

Kind regards,

Hamufare Dumisani Mugauri, Ph.D. Medicine and Health Sciences

Academic Editor

PLOS ONE
---

## [Author Response · Author response to Decision Letter 2]

11 Jul 2025

Dear Editor,

Thank you for the opportunity to improve the organization of our manuscript. Below are the response point-by-point to journal requirements and editor’s feedback. We hope our revised version meets your expectations.

Editor comments:

PLOS ONE considers qualitative and mixed-methods studies for publication. We recommend that authors use the COREQ checklist, or other relevant checklists listed by the Equator Network, such as the SRQR, to ensure complete reporting (http://journals.plos.org/plosone/s/submission-guidelines#loc-qualitative-research).

Response: We used the COREQ checklist to report methods and results of this manuscript. This is indicated in lines 117-118 and the checklist is included as S2 Table.

In general, we would expect qualitative studies to include the following:

1) defined objectives or research questions;

Response: Defined objectives are indicated in lines 107-110 and 121-124.

2) description of the sampling strategy, including rationale for the recruitment method, participant inclusion/exclusion criteria and the number of participants recruited;

Response: The “Selection of Participants” sub-section in the Methods describe sampling strategy, including rationale for the recruitment method, participant inclusion/exclusion criteria and the number of participants recruited. We added sub-headings to help reader identify the two level of participants (i.e., municipalities, and implementation teams) included in the study. In addition, as specified in COREQ, we added the method of approach to participant selection. These changes are indicated in lines 132-163.

3) detailed reporting of the data collection procedures;

Response: We reorganized the headings in the “Methods” section to clarify the sub-sections of the data collection procedures. This is indicated in lines 165-203.

4) data analysis procedures described in sufficient detail to enable replication;

Response: We reorganized the headings in the “Methods” section to clarify the sub-sections of the data analysis procedures. This is indicated in lines 205-257.

5) a discussion of potential sources of bias;

Response: Limitations and potential sources of bias are presented between lines 600-616.

6) a discussion of limitations.

Response: Limitations and potential sources of bias are presented between lines 600-616.

Journal Requirements:

Response: No changes in the reference list were made at this time.

---

## [Editor Report · Decision Letter 2]

Addressing Food Insecurity in Early Childhood Programs through a Health Equity Lens: A Qualitative Case Study of Brazil’s Criança Feliz Program.

PONE-D-24-37907R2

Dear Dr. Buccini,

We’re pleased to inform you that your manuscript has been judged scientifically suitable for publication and will be formally accepted for publication once it meets all outstanding technical requirements.

Kind regards,

Hamufare Dumisani Mugauri, Ph.D. Medicine and Health Sciences

Academic Editor

PLOS ONE
---

## [Editor Report · Acceptance letter]

PONE-D-24-37907R2

PLOS ONE

Dear Dr. Buccini,

I'm pleased to inform you that your manuscript has been deemed suitable for publication in PLOS ONE. Congratulations! Your manuscript is now being handed over to our production team.

Kind regards,

on behalf of

Dr Hamufare Dumisani Mugauri

Academic Editor

PLOS ONE